# Magneto-Structural Analysis of Hydroxido-Bridged Cu$^{II}_2$ Complexes: Density Functional Theory and Other Treatments

Debpriyo Goswami [ID], Shanti Gopal Patra *[ID] and Debashis Ray *

Department of Chemistry, Indian Institute of Technology, Kharagpur 721302, India; debpriyo0911@gmail.com
* Correspondence: patrashantigopal@gmail.com (S.G.P.); dray@chem.iitkgp.ac.in (D.R.)

**Abstract:** A selection of dimeric Cu(II) complexes with bidentate N,N′ ligands with the general formula [Cu(L)(X)($\mu$-OH)]$_2$·$n$H$_2$O and [Cu(L)($\mu$-OH)]$_2$X$_2$·$n$H$_2$O were magneto-structurally analyzed using the Density Functional Theory (DFT). A Broken Symmetry-Density Functional Theory (BS-DFT) study was undertaken for these complexes with relevant decomposition schemes that gave insight into the effect of the nature of the ligand and coordination environment on the DFT-predicted coupling constants ($J$). The impact of the spin population, which correlates well with the Cu-O-Cu bridging angles and the calculated coupling constant ($J$) values, was studied. The models were further refined using a complete active space self-consistent field (CASSCF) while expanding the active space from 2 orbitals 2 electrons (2,2) to 10 orbitals 18 electrons (18,10). These models were approximated using multireference methods (n-electron valence state perturbation theory and difference dedicated configuration interaction), and a better approximation of $J$ values was found as expected. Orbitals involved in the superexchange pathway were also visualized.

**Keywords:** magnetic coupling constant ($J$); superexchange; coupling constant decomposition; broken symmetry; complete active space (CAS); hydroxido-bridged; dinuclear copper complex

## 1. Introduction

Dinuclear copper complexes have been of significant interest as a class of compounds since the discovery of the various copper-containing metalloproteins, such as hemocyanin and tyrosine, involved in oxygen transport and metabolism [1,2]. The oxygenated form of hemocyanin and tyrosine, or oxyhemocyanin (oxyHc) and oxytyrosine (oxyTyr), respectively, were found to be a Cu$_2$($\mu$-$\eta^2$:$\eta^2$-O$_2$) system with the two copper atoms at the active site bridged by a peroxide ion. Insight into the structure and bonding of this system was made possible due to the synthesis of a biomimetic model by Kitajima et al. [3]. The model was further studied theoretically using the biomimetic model by Takano et al. to explain the nature of its magnetic coupling using superexchange [4,5].

The magnetically condensed dimers of first-row transition metals have been found to show a correlation between their structures and magnetic properties [6]. Dinuclear copper complexes are unique among these because of their potential applications as biomimetics of the copper active sites of coordination polymers and metalloproteins, as catalysts, and in magnetochemistry. The magnetic behavior of Cu(II) complexes is also one of the most straightforward systems to study because it is the smallest transition metal with a d$^9$ system. This also allows Cu(II) to form paramagnetic complexes in which the unpaired electrons can interact, resulting in magnetically condensed systems.

The complexes in this study all involve the Cu$_2$O$_2$ core with two Cu(II) atoms linked via two hydroxyl bridges. The study of hydroxy-bridged copper complexes with secondary and tertiary amines established the existence of such dimeric complexes [7–9]. In 1972, evidence of dimerization in bipyridyl copper nitrate was shown, which previous studies into copper bipyridyl systems had ignored, giving us another set of complexes to study [10].

The next decade saw a further investigation into these dimeric copper systems exploring the bonding and crystal structure in these systems [11–19]. Further magnetic study into those complexes followed [6,20–28]. A correlation between the Cu-O-Cu bridging angle and magnetic coupling constant (2*J*) was discovered by McGregor et al. [24], which was validated further by experimental data for similar complexes as they became available, which was later studied over multiple complexes and reported and interpreted [28,29]. Recently, these magnetic properties have been reviewed by various theoretical methods to validate past results and ascertain a proper approximation methodology [27,30–34].

Magnets based on molecular entities or single-molecule magnets (SMM) have been thoroughly explored experimentally [35–39]. Past work has also shown the capacity of such complexes to have tunable magnetic properties due to many possible structural distinctions [40]. Refining a theoretical methodology for approximating and predicting magnetic properties in these complexes will allow targeted experimental study.

For this theoretical study, a selection was made of a variety of dimeric copper complexes. These complexes were chosen based on factors such as ligands, counter anions, bonding, coordination environment, and others, taking into account the availability of the literature, experimental data, and crystallographic data. The final set of eight complexes presented includes (a) four chelating bidentate ligands (bipyridyl (bipy), dimethylaminoethylpyridyl (dmaep), ethylaminoethylpyridyl (eaep), and tetramethylethylenediamine (tmen)) and monodentate methyl-imidazole (miz); (b) three counter-anions/ligands (nitrate, perchlorate, bromide); (c) H-bonding (present or absent); (d) three types of coordination environments on Cu(II) (square planar, tetragonal pyramidal and distorted octahedral).

In this work, we report the results of the computational investigation of these complexes by using increasingly advanced and refined levels of theory. The complexes were first studied using a broken symmetry approach and then using multireference methods on a more sophisticated model by expanding the active space. These results were then used to corroborate magneto structural correlations discovered in previous studies and identify possible new correlations.

## 2. Computational Methodology

This study reports the results of different theoretical treatments on 8 dimeric copper complexes (Figures 1 and 2). The self consistent field (SCF), broken symmetry density functional theory (BS-DFT), complete active space (CASSCF), and approximations using n-electron valence perturbation theory (NEVPT2) and difference dedicated MRCI (DDCI3) were all performed on the ORCA 4.2.1 package [41]. Orbital visualizations were conducted on ORCA-enhanced Avogadro.

The complexes' structures from X-ray diffraction study were obtained from the relevant CCDC Identifiers, as listed in Table 1. The H atom locations reported in such X-ray diffraction studies are known to be unreliable and thus were optimized while locking the locations of the heavier atoms and the unconnected counter anions that were removed. Then, unrestricted Kohn–Sham DFT calculations were performed on the complexes with the Becke 3-parameter Lee–Yang–Parr (B3LYP) functional as it has been found to give the best results while keeping computational costs low. The complexes use a common hybrid basis set with a triple zeta valence potential (def2-TZVP) for the heavy copper atoms and the other atoms using a basis set with split valence potential (def2-SVP). The DFT results can be further improved by using higher basis sets.

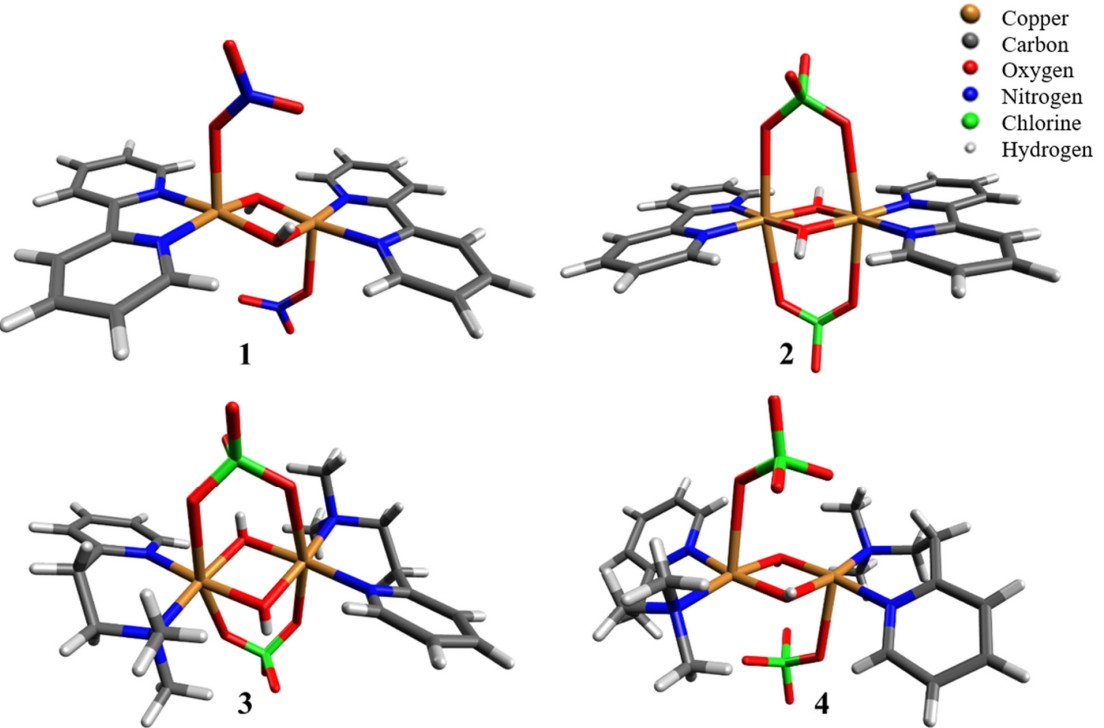

**Figure 1.** 3D molecular structures for complexes 1–4. (**1**) bisbipyridyl-μ-dihydroxidodicopper(II) dinitrate; (**2**) bisbipyridyl-μ-dihydroxodicopper(II) diperchlorate; (**3**) α-di-μ-hydroxo-bis[2-(2-dimethylaminoethyl) pyridinecopper(II)] diperchlorate; (**4**) β-di-μ-hydroxo-bis[2-(2-dimethylaminoethyl)pyridinecopper(II)] diperchlorate.

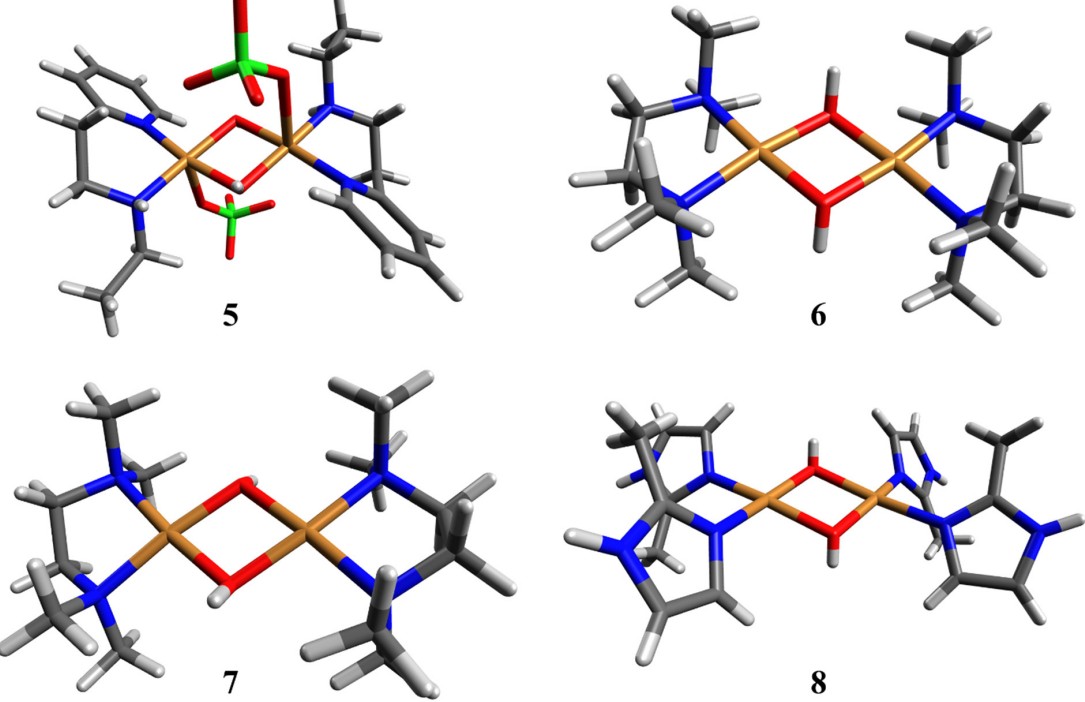

**Figure 2.** 3D molecular structures for complexes **5–8**. (**5**) di-μ-hydroxo-bis[2-(2-ethylaminoethyl)pyridine]dicopper(II) diperchlorate; (**6**) di-μ-hydroxo-bis(N,N,N′,N′-tetramethylethylenediamine) dicopper(II) perchlorate; (**7**) di-μ-hydroxobis(N,N,N′,N′-tetramethylethylenediamine)dicopper(II)bromide; (**8**) di-μ-hydroxybis[di(2-methylimidazole)copper(II)]diperchloratedihydrate. The atom coloring is the same as shown in Figure 1.

**Table 1.** Structural and magnetic properties of dimeric copper complexes.

| Complex | Chemical Formula | Cu-Cu (Å) | Cu-O (Å) | Cu-N (Å) | ∠Cu-O-Cu (deg) | 2J (cm$^{-1}$) | CCDC Identifier | Refs |
|---------|------------------|-----------|----------|----------|----------------|----------------|-----------------|------|
| 1 | [Cu(bipy)OH]$_2$(NO$_3$)$_2$ | 2.847 | 1.922 | 1.999 | 95.6 | 172 | 1,115,166 | [10,24] |
| 2 | [Cu(bipy)OH]$_2$(ClO$_4$)$_2$ | 2.871 | 1.919 | 1.990 | 96.9 | 93 | 1,010,002 | [11,28] |
| 3 | α-[Cu(dmaep)OH]$_2$(ClO$_4$)$_2$ | 2.938 | 1.942 | 2.037 | 98.4 | −4.8 | 1,132,368 | [12,26] |
| 4 | β-[Cu(dmaep)OH]$_2$(ClO$_4$)$_2$ | 2.935 | 1.910 | 2.035 | 100.4 | −201 | 1,297,137 | [13] |
| 5 | [Cu(eaep)OH]$_2$(ClO$_4$)$_2$ | 2.917 | 1.929 / 1.904 | 1.990 / 2.027 | 99.8 / 99.5 | −130 | 1,171,306 | [14,21] |
| 6 | [Cu(tmen)OH]$_2$(ClO$_4$)$_2$ | 2.966 | 1.929 | 2.014 | 101.6 | −360 | 1,178,533 | [9,15] |
| 7 | [Cu(tmen)OH]$_2$Br$_2$ | 3.000 | 1.902 | 2.030 | 104.1 | −509 | 1,177,680 | [16,22] |
| 8 | [Cu(2miz)OH]$_2$(ClO$_4$)$_2$·2H$_2$O | 2.988 | 1.959 | 1.977 | 99.4 | −175 | 1,177,096 | [17,20] |

The *J* decomposition scheme was obtained from the Kohn–Sham BSDFT calculations, where *J* can be decomposed using the Heisenberg–Dirac–van Vleck (HDvV) model, as mentioned in Equation (1). Here, *J* is represented as a sum of three different contributions: $J_0$ (direct exchange), $\Delta J_{KE}$ (kinetic exchange), and $\Delta J_{CP}$ (core polarization). While the direct exchange coupling is simply the difference in the energy levels of the triplet state and broken symmetry state with unoptimized orbitals, $\Delta J_{KE}$ is calculated by relaxing the magnetic orbitals in the lower spin state, allowing delocalization of magnetic orbitals in the broken symmetry state, and $\Delta J_{CP}$ uses relaxed core orbitals in both states.

$$J = J_0 + \Delta J_{KE} + \Delta J_{CP} \tag{1}$$

Ab initio multireference methods (such as CASSCF) use a systematically improvable approach and thus do not require the use of any semi-empirical procedures or parameters for calculating coupling constants. This approach estimates coupling constants in systems by directly calculating spin-state energies. The accuracy of the estimation depends upon the size of the basis set and the expansion of the active state [27,42]. To achieve numerical accuracy in this approach, static correlation effects are addressed using a density matrix renormalization group (DMRG) for an extensive active space. The remaining dynamic electron correlation effects are treated with a second-order perturbation theory such as NEVPT2 [27,32]. The methodology of CASSCF, followed by NEVPT2 by expanding the active space in several steps, allows one to study the nature of magnetic coupling where structural data is available and whether it is antiferromagnetic or ferromagnetic. While a flexible approach, this method requires observing the various orbitals and checking the combinations of suitable orbitals in active space to identify the proper pathway for the observed magnetic interaction. This also means that unless the path for a molecule can be estimated with reasonable accuracy for several systems, this methodology remains ineffective unless the experimental coupling constant (*J*) values are known. This would mean that expanding the active space for such molecules with no experimental data would be a waste of computational resources.

Approaches such as CASSCF that involve expanding the active space via permutation and combination is a brute-force approach that converges too slowly to be solely relied upon and is of practical utility for studying such dinuclear Cu(II) complexes. Difference-dedicated MRCI (DDCI) is a multireference configuration interaction (MRCI) method used for such centrosymmetric systems with more reliability and accuracy. For DDCI, the smallest active space of magnetic orbitals from CASSCF calculations is used as a reference. For these dinuclear Cu(II) systems, an active space consisting of two paramagnetic electrons distributed over two orbitals (CASSCF (2, 2)) is employed. The correlation energy is calculated variationally by considering different classes of excitation progressively. The excitations can be of several types: (i) hole excitation (h) from inactive to active orbitals, (ii) particle excitation (p) from active to virtual orbitals, and (iii) hole-particle excitation (h-p) from inactive to virtual orbitals. Depending on the number of electron excitations considered, the DDCI calculations can be denoted as DDCI1 for excitations involving 1h and 1p transitions (1h, 1p, 1h-1p); DDCI2 for excitations involving 2h and 2p transitions; and DDCI3 for (2h-1p) and (1h-2p) transitions. It is at the DDCI3 level that quantitative

predictions for magnetic coupling are expected, although the lower DDCI2 level calculations have occasionally been deemed satisfactory [9]. The two major conclusions are that DDCI2 is inadequate and that DDCI3 is necessary to obtain reliable results. The higher accuracy for DDCI3 approximations compared to DMRG-NEVPT2 calculations, even at a higher active space, indicates that electron excitations are an essential factor in estimating dynamic electron correlation for these dinuclear Cu(II) complexes. It should be noted that DDCI is not always applicable to different systems, such as non-centrosymmetric systems, as mentioned above, and higher nuclearity complexes, where DMRG-based methods, such as NEVPT2, have a broader scope and are considered generally applicable.

The CASSCF calculations were implemented with two configurations for active spaces ((2,2) and (18,10)). The notations here are written as (*n*,*m*), which denote an active space of *n* electrons over *m* orbitals. The starting orbitals for the CASSCF calculations were derived from the broken symmetry calculations using the B3LYP method. Subsequently, the NEVPT2 and DDCI3 (DDCI with 3 degrees of freedom) multireference methods were applied to these active spaces. The RI-JK approximation was used where applicable, with the def2-SVP basis set and a def2-JK auxiliary basis set. The visualization of orbitals was conducted using ORCA-enhanced Avogadro.

## 3. Results and Discussion

### 3.1. Structural Description of the Copper Complexes

The complexes are all of the general forms $[Cu(L)(X)(\mu\text{-}OH)]_2 \cdot nH_2O$ or $[Cu(L)(\mu\text{-}OH)]_2 X_2 \cdot nH_2O$, as shown in Figure 3 (the ChemDraw structures are given in Figure S1). Hydroxide ions bridge the two copper atoms with the four atoms falling in the same plane because of the centrosymmetry in these complexes, with some minor deviations in complex **5**. Here, L represents the bidentate ligands that coordinate via two nitrogen atoms, with the exception of complex **8**, which has two monodentate 2-methylimidazole ligands resulting in the final complex being of the form $[Cu(L)_2(\mu\text{-}OH)]_2 X_2 \cdot nH_2O$ instead. The X represents the counter-anion that exists as a coordinated species in the first representation and uncoordinated in the latter. A comparison of various bond lengths and bond angles is given in Table S1 (also see Table 1). The Cu-Cu distance decreases as the ∠CuOCu decreases. The Cu-O and Cu-N bond lengths remain almost the same in all the complexes. Thus, the change in the $Cu_2O_2$ diamond core is responsible for the difference in magnetic exchange coupling.

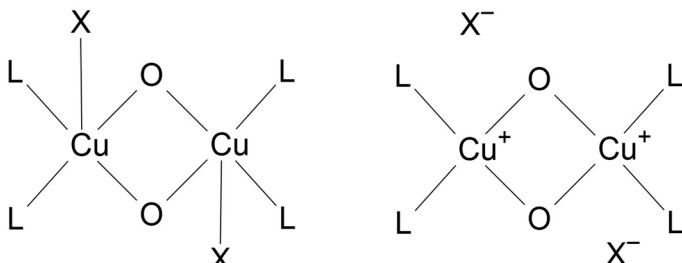

**Figure 3.** Dinuclear Cu(II) complex with coordinated and uncoordinated anions.

Complexes **1** and **2** have a chelating bipyridyl (bipy) ligand; complexes **3**, **4**, and **5** have derivative aminoethylpyridyl (aep) ligands; complexes **6** and **7** have a tetramethylethylene-diamine (tmen) ligand; and complex **8** has a 2-methylimidazole ligand. Complex **1** has a nitrato, complex **7** has a bromide, and all other complexes have perchlorate counter-anions, some of which exist as uncoordinated species, as described further ahead.

The coordination geometry around copper in these species also varies. Complexes **6** and **7** have a square planar geometry with the Cu(II) center cis-coordinated to the two oxygen atoms from the hydroxo-bridges and two nitrogen atoms from the chelating ligand [15,16]. Complexes **1**, **4**, and **5** have a distorted tetragonal pyramid, with the fifth axial site occupied by coordinated oxygen from the nitrate and perchlorate anion [10,13,14]. In complexes **4** and **5**, the perchlorate anion was found to be weakly coordinated compared

to the nitrate in complex **1**; based on the Cu-O separation in complex **4,** this was deemed to be semi-coordination as described by Hathaway et al. [43]. The copper atom in such complexes with tetragonal pyramidal geometry tends to rise above the basal plane with the two nitrogen (ligand) and two oxygen (bridging) ligands and toward the fifth coordinated atom. This copper atom is found to be almost in the basal plane, which is a unique feature in a departure from other copper dimers with tetragonal pyramidal geometry. The substituted animoethylpyridyl ligands in complexes **4** and **5** partially prevent the other perchlorate ion from approaching the copper atom from the opposite side [13].

Complexes **2**, **3**, and **8** have a trans-distorted octahedral (4 + 2) geometry, with the two nitrogen and bridging oxygen atoms taking the equatorial positions and the two perchlorate ions weakly bound at the axial positions [11,12,17]. The perchlorate ions also act as bridging ligands and may be considered semi-coordinated due to the weak interaction. In the DMAEP complexes **3** and **4**, the sixth coordination site of the α complex is not inhibited. The complexes all have a core with the adjoining geometry on the two copper centers with a common edge and the eight equatorial atoms almost coplanar. In the case of the distorted tetragonal pyramids, the pyramids face opposite directions.

The complexes also vary in hydrogen bonding, with complexes **2**, **4**, **5,** and **6** showing no hydrogen bonding or very insignificant interactions based on van der Waals radii and solid-state infrared spectrum. Complex **3** shows intramolecular hydrogen bonding show intramolecular hydrogen bonding where a third oxygen atom other than the ones coordinated to copper is hydrogen-bonded to the bridged hydroxyl oxygen. Complex **1** and **8** show intermolecular hydrogen bonds with nitrate ion hydrogen-bonded to the hydroxyl oxygen of the adjacent dimer for complex **1** and the water of crystallization linking two dimers via the hydroxyl oxygen and the perchlorate oxygen of an adjacent dimer for the 2-methylimidazole complex **8**. Complex **7** has hydrogen-bonded bromide ions to the nearest bridged hydroxyl oxygen.

The crystals of most complexes are monoclinic (a $\neq$ b, a $\neq$ c, α = β = 90°, γ $\neq$ 90°) except complexes **1**, **3**, and **7** (Table S2). The α-DMAEP complex **3** is triclinic, and the tetramethylethylenediamine bromide complex **7** is orthorhombic. The bipyridyl nitrate complex **1** is a non-rhombohedral body centered with α = 91.58°, β = 97.49°, γ = 90.36° which can be considered a distorted monoclinic form with two angles between edges slightly deviating from the required 90°.

*3.2. Magneto-Structural Correlation with Broken Symmetry DFT*

The strength of the coupling interaction in molecules with multiple magnetic centers is given by a magnetic coupling constant (*J*). This magnetic coupling constant can be determined experimentally by measuring the temperature variance of magnetic susceptibility and fitting the data into the van Vleck equation [24]. Furthermore, 2*J* values obtained using this are known to be insensitive to large positive values of *J* and, as is the case with such estimation, using a Broken Symmetry approach using the hybrid B3LYP functional, which tends to give a larger positive value, thus results in overestimation coupling constants in ferromagnetic molecules and wrong identification of weakly antiferromagnetic compounds as ferromagnetic [44]. Overestimation of ferromagnetic interaction causes this skew in an approximation of *J* values due to the higher calculated values of the singlet state energy. The broken symmetry DFT approximations are reported in Table 2. The approximations yield an $R^2$ value of 89.2 (6)% upon calculating the variance of approximated values of 2*J* compared to experimental determinations. This shows the basis sets skew the data but not so heavily as to impact further correlation analysis. The DFT calculations reported are expected to become more accurate upon using higher basis sets and more accurate functionals, which require higher computing resources and time.

**Table 2.** The 2*J* values obtained from experimental data from past studies and Broken Symmetry DFT approximations.

| Complex | Chemical Formula | 2J (Exp) (cm$^{-1}$) | 2J (BS-DFT) (cm$^{-1}$) |
|---|---|---|---|
| **1** | $[Cu(bipy)OH]_2(NO_3)_2$ | 172 | 335.34 |
| **2** | $[Cu(bipy)OH]_2(ClO_4)_2$ | 93 | 127.88 |
| **3** | $\alpha$-$[Cu(dmaep)OH]_2(ClO_4)_2$ | −4.8 | 166.22 |
| **4** | $\beta$-$[Cu(dmaep)OH]_2(ClO_4)_2$ | −201 | −99.10 |
| **5** | $[Cu(eaep)OH]_2(ClO_4)_2$ | −130 | 56.78 |
| **6** | $[Cu(tmen)OH]_2(ClO_4)_2$ | −360 | −77.72 |
| **7** | $[Cu(tmen)OH]_2Br_2$ | −509 | −341.04 |
| **8** | $[Cu(2miz)OH]_2(ClO_4)_2 \cdot 2H_2O$ | −175 | −85.50 |

The linear correlation between the ∠Cu-O-Cu bridging angle has been well-established in past studies. Here, we were able to see the correlation between the bridging angle φ and the coupling constant (both experimental and theoretical. A strong correlation ($R^2$ = 99.82%) is also observed between the Cu-Cu separation and the coupling constants obtained experimentally. This linear relation has two outliers in complexes **3** and **8,** as seen in Figure 4. While complex **3** significantly differs from the rest in crystal packing (triclinic), the 2-methylimidazole complex (**8**) has a non-chelating monodentate ligand.

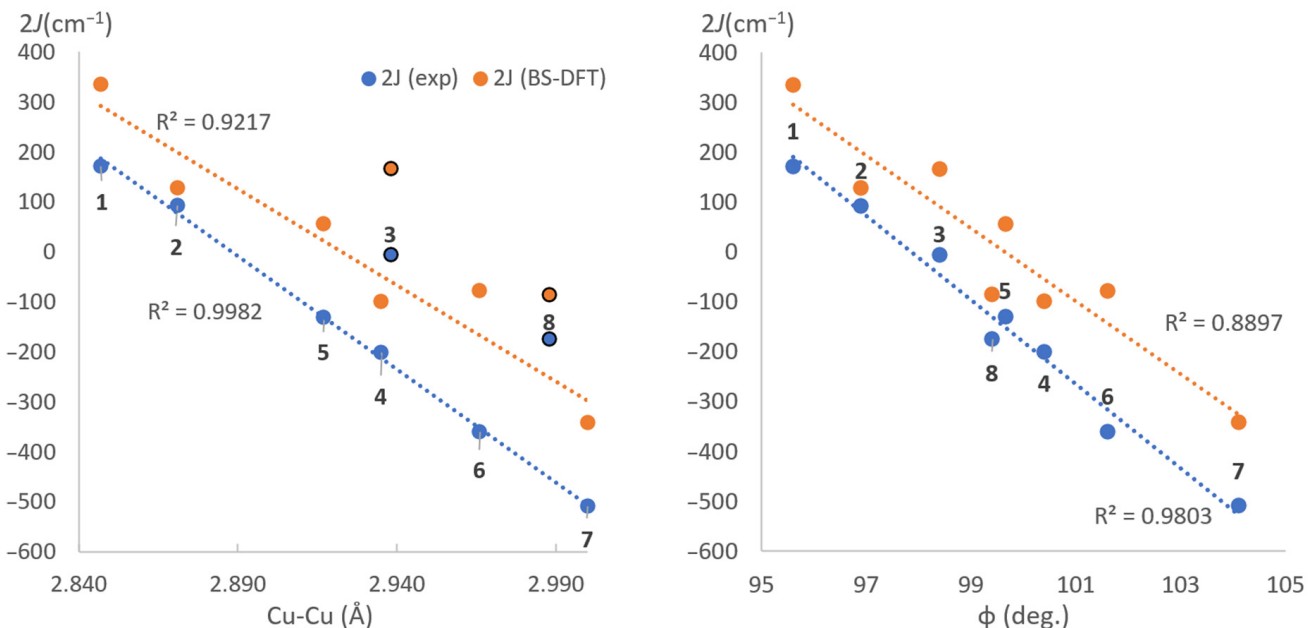

**Figure 4.** Magneto-structural correlation between experimental and theoretical coupling constant values, and Cu-Cu separation (**left**) and Cu-O-Cu angle (**right**) with the data points labelled with their respective complex numbers.

### 3.3. Decomposition Scheme for Broken Symmetry DFT Formulation

The BS-DFT calculations also provide a set of decomposition values for the magnetic coupling constant, splitting the superexchange coupling constant into a direct exchange $J_0$, a kinetic exchange $\Delta J_{KE}$, and a core polarisation $\Delta J_{CP}$ reported in Table 3.

**Table 3.** Decomposition scheme from BS-DFT approximation calculations.

| Complex | CCDC No. [a] | 2*J* (Exp) | ∠OH-Cu$_2$O$_2$ | *J*$_0$ | Δ*J*$_{KE}$ | Δ*J*$_{CP}$ |
|---|---|---|---|---|---|---|
| 1 | BPYHCU | 172 | 126 | 132.77 | −31.39 | 25.77 |
| 2 | DPCUCL15 | 93 | 146.5 | 144.69 | −136.89 | 25.89 |
| 3 | CUDMAP | −4.8 | 117.3 | 122.66 | −84.98 | 42.84 |
| 4 | XMPYCU | −201 | 139 | 134.85 | −217.97 | 37.94 |
| 5 | HAEPCU10 | −130 | 125 | 125.53 | −186.91 | 58.22 |
| 6 | HTMCUP | −306 | [b] | 125.52 | −229.79 | 23.34 |
| 7 | HOMECU10 | −509 | [b] | 124.69 | −308.73 | 17.87 |
| 8 | HMIMCU10 | −175 | 124.5 | 135.54 | −244.8 | 35.46 |

[a] Cambridge Crystallographic Data Centre (CCDC). [b] Missing data due to unreported positions in crystallographic studies due to inaccuracies from the references in Table 1.

The direct exchange $J_0$ exhibits a consistent value across the entire range, ranging from 122.66 to 144.69 cm$^{-1}$. This similarity arises from a shared characteristic among these systems: the hydroxide bridge, which functions as a non-magnetic intermediate atom for superexchange interactions. The angle that the O-H bond of the bridging hydroxido group makes with the core Cu$_2$O$_2$ plane is found to have a strong correlation with the direct exchange contribution ($R^2$ = 73.6%), as seen in Figure 5. The higher the angle, the more negative the exchange coupling. Thus, when hydrogen is out of the plane, oxygen p orbitals are more involved in the superexchange process.

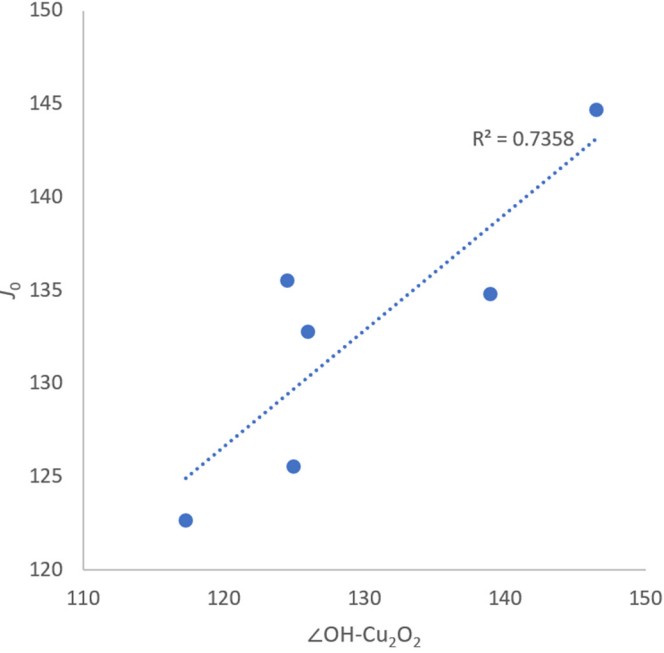

**Figure 5.** ∠OH-Cu$_2$O$_2$ tilt angle vs. Direct exchange contribution to coupling constant.

The kinetic exchange contribution Δ$J_{KE}$ changes significantly from one complex to the other with a change in the ligands. When observing the kinetic exchange Δ$J_{KE}$ values for the complexes, it is evident that these values undergo significant changes with variations in the counter anion. For example, on moving from nitrate to perchlorate in complexes **1** and **2** with the same neutral ligand (bipyridyl), it changes from −31.39 to −131.89 cm$^{-1}$ where the core polarisation Δ$J_{CP}$ remains very close (25.77, 25.89). Complexes **6** and **7** have comparable values for core polarization contribution Δ$J_{CP}$; however, the kinetic exchange contribution Δ$J_{KE}$ changes significantly. Since they are structurally identical, the hydrogen bonding and proximity of the anion (if present as a distinguishing factor) could be the reason for such changes. While hydrogen bonding was negligible, if not absent in complex **6**, a strong OH stretching band at around 3410 cm$^{-1}$ confirms the presence of H-bonding

in complex **7** [16]. The effect of separating the counter-anion needs to be explored with other sets of complexes. Another observation was made with molecules **3**, **4,** and **5,** where a change in kinetic exchange contribution $\Delta J_{KE}$ was observed in complexes **3** and **4** with similar composition except for the coordination environment for the Cu(II) centers. The triclinic $\alpha$ form (complex **3**), belonging to a different structural class than the monoclinic $\beta$ form (complex **4**), has a higher coordination number for the Cu(II) center. Thus, it can be concluded that the coordination environment, precisely the coordination number here, affected kinetic exchange $\Delta J_{KE}$ with the need for further exploration. Comparison between complexes **4** and **5** shows a difference in $\Delta J_{KE}$ but not as high a difference as complexes **1** and **2** due to the two different ligands, which differ only by the groups attached to the coordinating atom N, thus varying only due to inductive effect and the s-character of the orbital with the lone pair.

The kinetic exchange contribution was found to be the strongest factor affecting the coupling constant as the ligand and counter-anion are changed. The direct exchange contribution changes affect the coupling constant weakly due to the common bridging hydroxido group but are found to reliably increase in value as the orientation of the hydroxide group changes concerning the $Cu_2O_2$ plane. The core polarization contribution is negligible but correlated to the O-H bond length of the bridging hydroxido group.

### 3.4. Mulliken Charge Density and Spin Population Analysis

The BS-DFT calculations performed on ORCA also give us a Mulliken charge density and spin population set. Upon analyzing the data presented in Table 4, it was observed that not all molecules exhibit a triplet or singlet ground state. Instead, an average anti-ferromagnetic or ferromagnetic interaction was found, aligning with previous theoretical propositions made by Noodleman [45,46]. Instead, the ground state is a mix of molecules in the singlet or triplet ground state. The spin population on one of the Cu centers strongly correlates to the coupling constant (Figure 6, $R^2$ = 83.4%). The lower the spin population, the higher the antiferromagnetic coupling. Obviously, in the antiferromagnetically coupled state, the spin of the two metal centers is oppositely coupled, and increased covalency leads to a decrease in spin population. Small values of the spin population suggest a strong delocalization (M-L covalency) of the two copper centers.

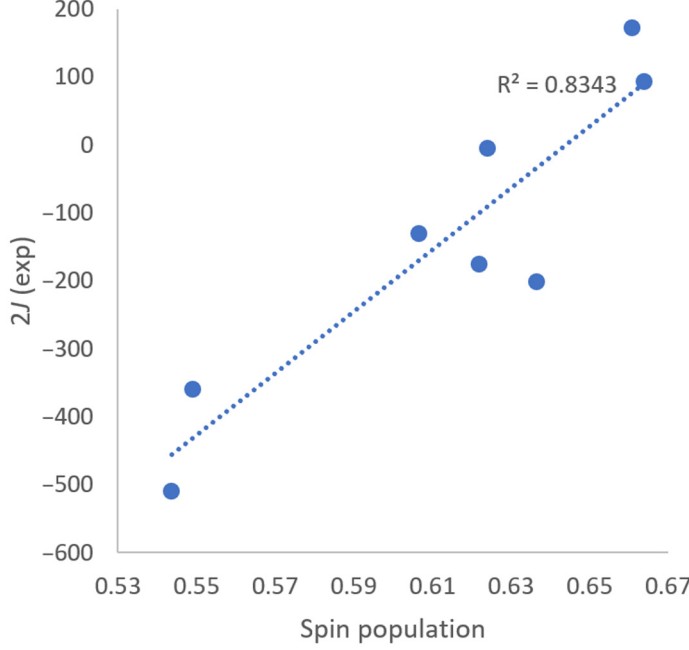

**Figure 6.** Spin population vs. coupling constant.

**Table 4.** Charge density and spin population on one of the Cu centers and $<S^2>$ values from DFT calculations.

| Complex | 2J (Exp) | Partial Charge | Spin Population | $<S^2>_{HS}$ | $<S^2>_{BS}$ |
|---|---|---|---|---|---|
| **1** | 172 | 0.3798 | 0.660777 | 2.0051 | 1.0027 |
| **2** | 93 | 0.3591 | 0.663756 | 2.0046 | 0.9961 |
| **3** | −4.8 | 0.3227 | 0.623951 | 2.0048 | 0.9982 |
| **4** | −201 | 0.3685 | 0.636479 | 2.0046 | 0.9901 |
| **5** | −130 | 0.3006 | 0.606567 | 2.0052 | 0.9907 |
| **6** | −360 | 0.2833 | 0.548932 | 2.0058 | 0.9868 |
| **7** | −509 | 0.2907 | 0.543658 | 2.0058 | 0.9798 |
| **8** | −175 | 0.3512 | 0.621797 | 2.0049 | 0.9877 |

The values of $<S^2>$ at the high spin state are close to the expected value of 2 according to S(S + 1), with S = 1 confirming negligible spin contamination at the high spin state. The computed values of $<S^2>$ at the broken symmetry state lie between the value of spin at the broken symmetry singlet state (S = 0) and $<S^2>$ = 2, as is expected [47].

The spin density visualization in Figure 7 shows us the lower spin (Broken Symmetry) state of complex **1**. We see how the spin on one copper center is opposite to the spin on the other copper center.

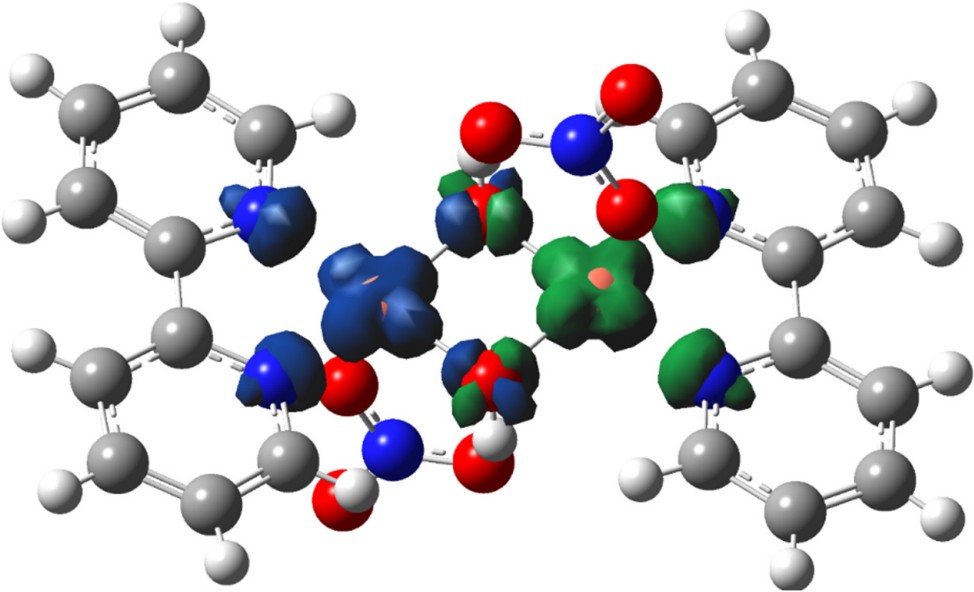

**Figure 7.** Spin density visualization for complex **1** in the broken symmetry state. The blue color surface represents the positive spin, and the green color surface represents the negative spin. Atom coloring: blue, N; grey, C; white, H; red, O; orange, Cu.

### 3.5. Complete Active Space SCF and N-Electron Valence Perturbation Theory

The next level of the theory involves defining an active space. We start with a smaller active space of the highest energy singly occupied orbitals. This gives us an active space of two orbitals with two electrons, followed by multireference calculations NEVPT2 and DDCI3. The magnetic orbitals from CAS (2, 2) are visualized in Figure 8. With the NEVPT2 calculations with a smaller active space, we notice that it manages to be closer to the experimental values of coupling constant by up to 9.37% and does better at identifying more complexes correctly as ferromagnetic/antiferromagnetic (Table 5).

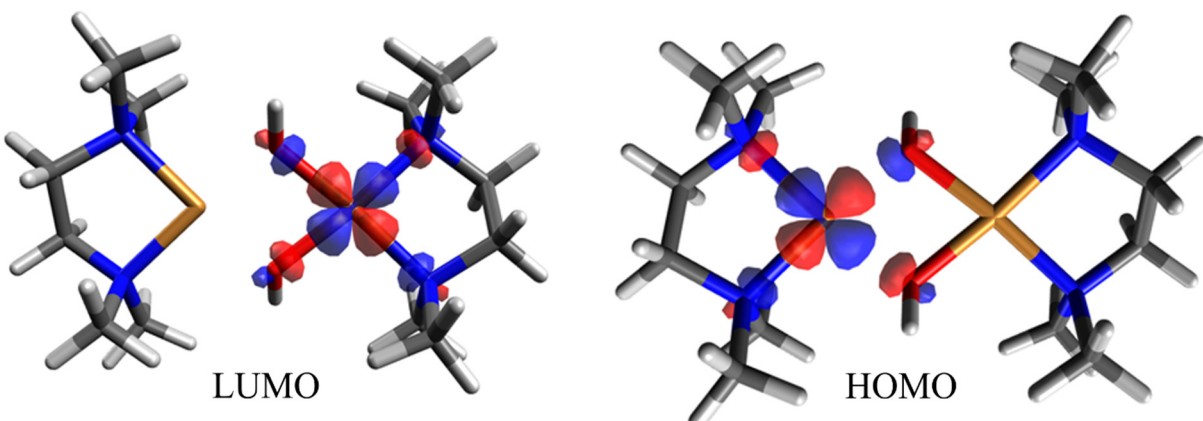

**Figure 8.** Orbital visualizations for CAS-SCF implementation with 2 electrons over 2 orbitals and its NEVPT2 approximation. Atom coloring: blue, N; grey, C; white, H; red, O; orange, Cu. Blue and red isosurface represents the positive and negative sign of the orbital.

**Table 5.** CAS-SCF implementation with 2 electrons over 2 orbitals and its NEVPT2 approximation.

| Complex | 2J (Exp) | CAS (2,2) | NEVPT2 | NEVPT2 Correction |
|---|---|---|---|---|
| **1** | 172 | 31.2 | 35 | 2.70% |
| **2** | 93 | 33.4 | 33.6 | 0.34% |
| **3** | −4.8 | 29.8 | 29.7 | 0.29% |
| **4** | −201 | 26.7 | 18.9 | 3.43% |
| **5** | −130 | 21.4 | 8.1 | 8.78% |
| **6** | −360 | 4.9 | −22.5 | 7.51% |
| **7** | −509 | −22.4 | −68 | 9.37% |
| **8** | −175 | 15.5 | 3.7 | 6.19% |

After completing the initial calculations with an active space of two electrons distributed over two orbitals, the subsequent step involves gradually incorporating additional relevant orbitals into the active space. This iterative process allows for a progressive expansion of the active space. In the first iteration, the highest doubly occupied d orbitals of the metal atom are included as the initial set of orbitals in the active space. This selection forms the foundation for expanding the active space in subsequent iterations. The ORCA output file for the CAS (2, 2) calculation was analyzed to examine orbital contributions and identify occupied orbitals with lower energies than the HOMO-LUMO orbitals, with a significant contribution from the copper 3d orbitals. These orbitals were visualized to confirm d-type orbitals. The 8 selected orbitals are then moved out to the active space with the rotate features, giving a new active space with 10 orbitals (2 + 8) and 18 electrons (2 + 2 × 8). An additional set of CASSCF and NEVPT2 calculations was carried out, this time utilizing a larger active space of (18, 10). The purpose was to examine whether this expanded active space would yield improved agreement with experimental values compared to the smaller scope CAS calculations. By analyzing the results, it was expected to observe a general trend of values that align more closely with the experimental data. The values from the expanded CAS (18, 10) are reported in Table 6. The multireference NEVPT2 approximations on the expanded active space give a higher rate of recovery of the canonical coupling constant values at 10.09%. The negative values here indicate a need to choose another set of orbitals for the active space. The accuracy of all post-CASSCF methods is compared in Figure 9. The resultant CASSCF (18, 10) orbitals are visualized in Figure 10. It is observed that barring a few exceptions where an alternative active space would have to be explored, the values are closer to the experimental values as we move from CASSCF to NEVPT2 and expand the active space from (2, 2) to (18, 10). Figure 9 compares the theoretical values calculated compared to the experimental value and the accuracy visualized compared to the experimental value at 100%. Negative values indicate that the method was unable to

identify the ferromagnetic or antiferromagnetic coupling correctly. This can be rectified by calculations using a higher basis set for copper and expanding the active space further.

**Table 6.** CAS-SCF implementation with 18 electrons over 10 orbitals and its NEVPT2 approximation.

| Complex | 2J Exp | CAS (18,10) | NEVPT2 | NEVPT2 Correction |
|---|---|---|---|---|
| **1** | 172 | 31.1 | 41.1 | 7.03% |
| **2** | 93 | 33.2 | 34.3 | 1.51% |
| **3** | −4.8 | 25.4 | 32.8 | −8.67% |
| **4** | −201 | 23.6 | 41.1 | −6.32% |
| **5** | −130 | 17.4 | 10.2 | 7.40% |
| **6** | −360 | 4.8 | −28.1 | 9.04% |
| **7** | −509 | −22.5 | −71.5 | 10.09% |
| **8** | −175 | 12.5 | 8.8 | 3.52% |

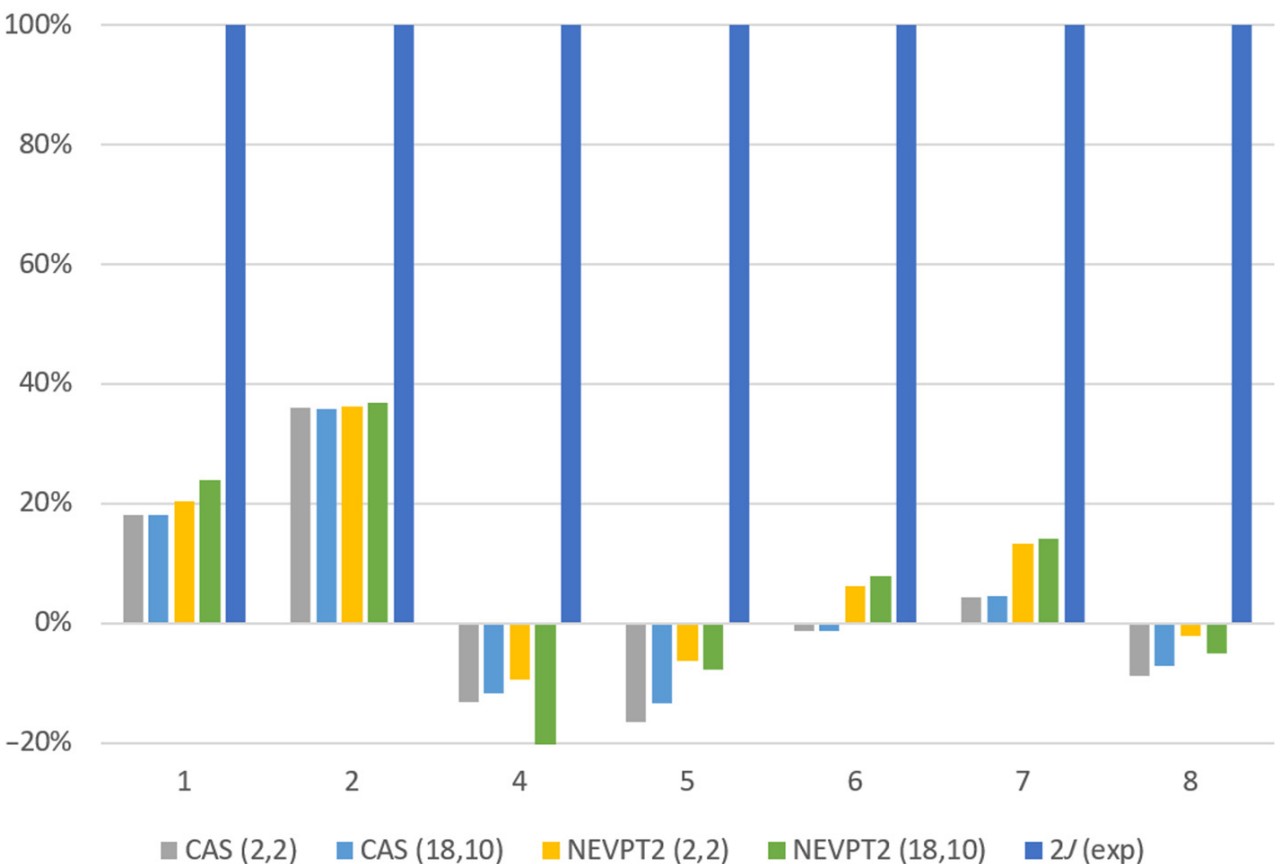

**Figure 9.** Comparison of approximations at different levels of theory (CASSCF and NEVPT2).

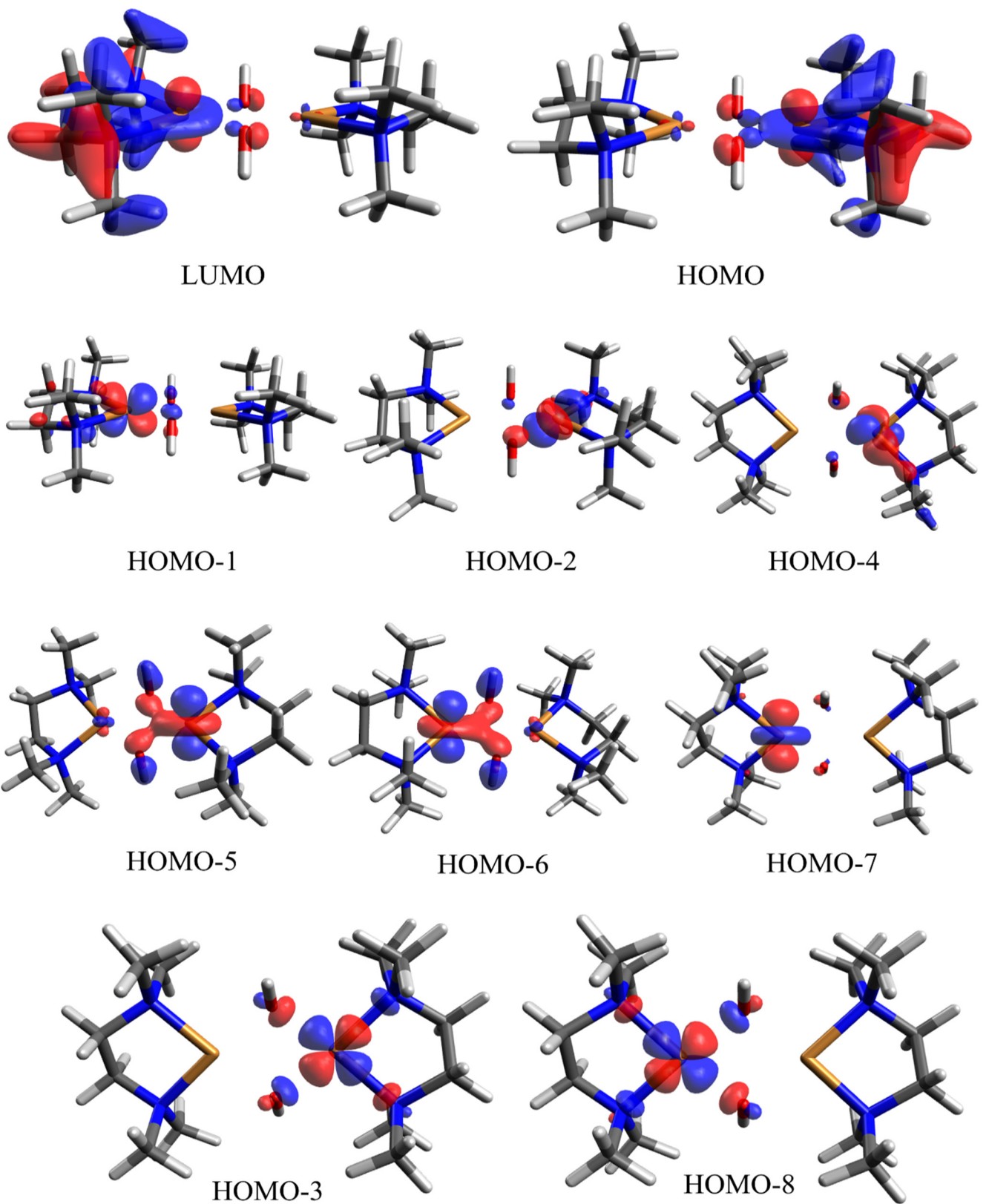

**Figure 10.** Orbital visualizations for CAS-SCF implementation with 18 electrons over 10 orbitals and its NEVPT2 approximation. Atom coloring: blue, N; grey, C; white, H; red, O; orange, Cu. Blue and red isosurface represents the positive and negative sign of the orbital.

### 3.6. Difference Dedicated MRCI

While computational and resource limits did not allow for all DDCI3 calculations, the values generated give us a good insight into the vastly increased accuracy of the method, even at a smaller active space. The DDCI3 approximations over the (2,2) active space yield far better results recovering 19.95% and 38.51% compared to the NEVPT2 approximations over the (18,10) active space as reported in Table 7. Figures 9 and 11 show how the values approach the experimental data as the complexity of the method increases.

**Table 7.** DDCI3 implementation and NEVPT2 approximation.

| Complex | 2J Exp | NEVPT2 | DDCI3 | DDCI3 Correction |
|---------|--------|--------|-------|------------------|
| **6** | $-360$ | $-28.1$ | $-94.3$ | 19.95% |
| **7** | $-509$ | $-71.5$ | $-240$ | 38.51% |

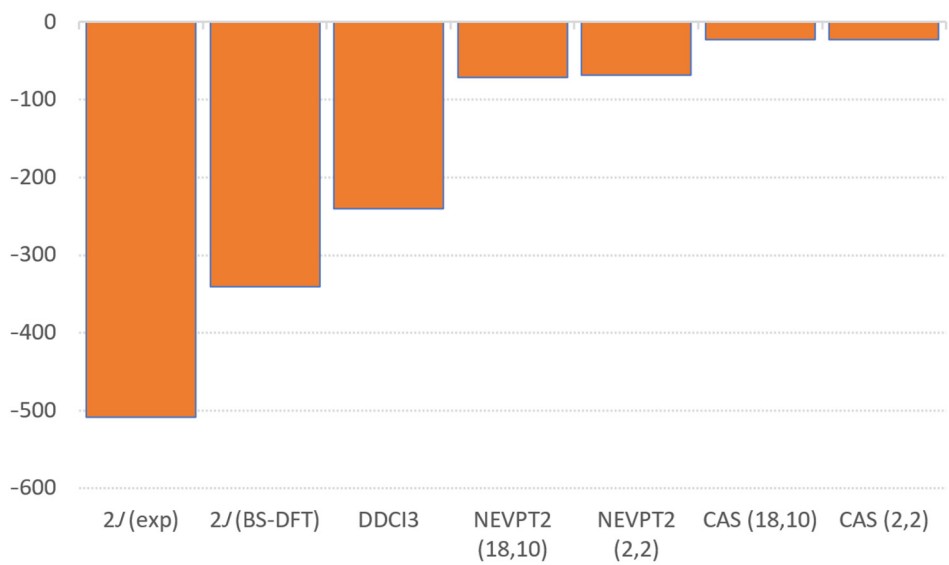

**Figure 11.** Coupling constant values from the literature and various levels of theory compared for $[Cu(tmen)OH]_2Br_2$.

### 4. Conclusions

The hydroxido-bridged dinuclear Cu(II) complexes studied here with theoretical methods verify the relation between the nature of the coupling between the two magnetic centers via superexchange with the bridging angle Cu-O-Cu. It shows the change from ferromagnetic to antiferromagnetic coupling as the bridging angle changes from 90° to a higher value. The calculated values correlate well with the experimental data. A plot of bridging angle vs. calculated $J$ gives a nice straight line with an $R^2$ of 0.87. By studying the decomposition scheme of the magnetic coupling constant, we see the direct exchange contribution to the coupling constant was found to be affected by the orientation of the O-H bond of the bridging group in relation to the planar $Cu_2O_2$ core. The kinetic energy contribution to the coupling constant is observed to be very significantly affected by the ligand and counter-anion attached. With the spin population study from the broken symmetry formulation, we see how the ground spin state of the complex is not always a singlet or a triplet but an intermediate state statistically. A correlation between the spin population and the bridging angle was also observed. The lower the spin population, the higher the antiferromagnetic coupling.

From the CASSCF and multireference methods of calculations, we systematically observed that the theoretically estimated values of magnetic coupling constants approach the experimental values as the active state is expanded from (2,2) to (18,10). NEVPT2 approximations lead to a higher accuracy than CAS-SCF methods. Even more accurate

results are obtained with the DDCI3 method recovering the coupling constant at a higher rather than previous levels of theory. The orbital visualizations also observed the magnetic orbitals involved and the superexchange pathway. This work highlights a pathway to better approximation via more resource-intensive options such as upgrading basis sets, further expanding active space, and applying multireference methods.

**Supplementary Materials:** The following supporting information can be downloaded at: https://www.mdpi.com/article/10.3390/magnetochemistry9060154/s1. Figure S1: Molecular structures for Complex 1-8.; Table S1: Structural data from X-Ray crystallographic study; Table S2: Unit Cell data from Crystallographic study.

**Author Contributions:** Conceptualization, D.G., S.G.P. and D.R.; methodology, D.G. and S.G.P.; software, D.G. and S.G.P.; validation, D.G., S.G.P. and D.R.; formal analysis, D.G. and S.G.P.; investigation, D.G. and S.G.P.; resources, D.G., S.G.P. and D.R.; data curation, D.G.; writing—original draft preparation, D.G. and S.G.P.; writing—review and editing, D.G., S.G.P. and D.R.; visualization, D.G. and S.G.P.; supervision, D.R.; project administration, D.R. All authors have read and agreed to the published version of the manuscript.

**Funding:** This research received no external funding.

**Institutional Review Board Statement:** Not applicable.

**Data Availability Statement:** Not applicable.

**Acknowledgments:** We acknowledge the National Supercomputing Mission (NSM) for providing computing resources of 'PARAM Shakti' at IIT Kharagpur, which is implemented by C-DAC and supported by the Ministry of Electronics and Information Technology (MeitY) and the Department of Science and Technology (DST), Government of India. S.G.P. thanks IIT Kharagpur for a postdoctoral fellowship.

**Conflicts of Interest:** The authors declare no conflict of interest.

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
