# Peer review of "Magneto-Structural Analysis of Hydroxido-Bridged CuII2 Complexes: Density Functional Theory and Other Treatments"

_magnetochemistry, doi:10.3390/magnetochemistry9060154_

Round 1
Reviewer 1 Report
In this manuscript, A selection of dimeric Cu(II) complexes with bidentate N, N’ ligands with general formula [Cu(L)(X)(μ-OH)]2⸳nH2O or [Cu(L)(μ-OH)]2X2⸳nH2O were magneto-structurally analysed on the Density Functional Theory (DFT) level. A Broken Symmetry-DFT (BS-DFT) study was undertaken for these complexes with relevant decomposition schemes. Here are some comments for this manuscript.
1. Introduction part should be improved to show the significance of this work.
2. The authors should review the manuscript carefully as there are many minor mistakes.
a. Cu(II) and copper(II) are used simultaneously.
b. Lots of abbreviations are used in this manuscript. The authors should make sure that the full name of these abbreviations is presented when they first appear. e.g., B3LYP.
c. For some references, [] is not used. e.g., refs 15, 16, 10, 13, 14, 11, 12, and 17 in Page 6, 2nd paragraph.
d. () is used for complexes (3), (1) and (7) in Page 7, 1st paragraph.
e. R2 should be R2. What is Jab in Page 7, 2nd paragraph?
3. There is no figure name for LUMO-HOMO in Page 13. Where is Figure 18, 19 in Page 14, 1st paragraph?
4. Figure 3 should be Cu-Cu separation (left) and Cu-O-Cu angle (right). Besides, please label the complexes in Figure 3.
5. What is the difference between complexes in Tables 1, 2 and Molecules in Table 3-7? Please present the structure of molecules if possible.
Minor polishing of English language required
Author Response
Reviewer 1
In this manuscript, A selection of dimeric Cu(II) complexes with bidentate N, N’ ligands with general formula [Cu(L)(X)(μ-OH)]2⸳nH2O or [Cu(L)(μ-OH)]2X2⸳nH2O were magneto-structurally analysed on the Density Functional Theory (DFT) level. A Broken Symmetry-DFT (BS-DFT) study was undertaken for these complexes with relevant decomposition schemes. Here are some comments for this manuscript.
- Introduction part should be improved to show the significance of this work.
- Please check the highlighted portion in the Introduction section on page 2 for the added portion that shows the significance of this work.
- The authors should review the manuscript carefully as there are many minor mistakes.
- Cu(II) and copper(II) are used simultaneously.
- Lots of abbreviations are used in this manuscript. The authors should make sure that the full name of these abbreviations is presented when they first appear. e.g., B3LYP.
- For some references, [] is not used. e.g., refs 15, 16, 10, 13, 14, 11, 12, and 17 in Page 6, 2 nd paragraph.
- () is used for complexes (3), (1) and (7) in Page 7, 1 st paragraph.
- R2 should be R 2 . What is Jab in Page 7, 2 nd paragraph?
- Inconsistencies and mistakes pointed out and similar ones have been fixed. Acronyms have been added at the point of first reference. The “Jab” in point e refers to coupling constant resulting from the two magnetic centres but has been changed to “J” for consistency with the rest of the work and to avoid any possible confusion.
- There is no figure name for LUMO-HOMO in Page 13. Where is Figure 18, 19 in Page 14, 1 st paragraph?
- The figure pointed out has been numbered. Figure 18, 19 were erroneously numbered according to a previous draft of the manuscript and has been correct to the updated numbering.
- Figure 3 should be Cu-Cu separation (left) and Cu-O-Cu angle (right). Besides, please label the complexes in Figure 3.
- The error here has been corrected.
- What is the difference between complexes in Tables 1, 2 and Molecules in Table 3-7? Please present the structure of molecules if possible.
- The complexes are the same and their CCDC notations have been used, the change in notation has been clarified at the point of introduction.
Comments on the Quality of English Language: Minor polishing of English language required
- Corrected.

Reviewer 2 Report
This paper reports on the computer calculations for hydroxido-bridged dinuclear copper(II) complexes.
There is a big gap between what the author is working on and the description of SMM in the introduction. In addition, the introduction does not state what exactly is to be clarified and to what extent.
The results does not have the author's own experimental data to guarantee the calculation results, nor does it give results that exceed the previously reported good correlation between the Cu-O-Cu angles and the magnetic interactions. Regarding the relationship between the angle formed by the O-H and Cu2O2 plane and the magnetic interaction, which was newly presented this time, the position of the hydrogen atoms is based on calculations, so there is little evidence to say that it is a new discovery. The other mentioned correlation between the 2J value and the calculated result was judged to have no chemical basis to support it.
Judging these comprehensively, this paper does not deserve to be published in Magnetochemistry.
Author Response
Reviewer 2
This paper reports on the computer calculations for hydroxido-bridged dinuclear copper(II) complexes.
There is a big gap between what the author is working on and the description of SMM in the introduction.
- The brief introduction and references to SMM has been added to serve as an entrypoint to look into molecular magnetism and how it relates to electronic structure.
In addition, the introduction does not state what exactly is to be clarified and to what extent.
- A section at the end of introduction has been added and and highlighted (page 2)
The results does not have the author’s own experimental data to guarantee the calculation results, nor does it give results that exceed the previously reported good correlation between the Cu-O-Cu angles and the magnetic interactions.
- This work is a theoretical study on several complexes with available experimental and crystallographic data which have been referenced in Table 1. The results while less accurate than past studies due to the use of functionals and basis sets that were more resource effecient, have been reported to serve as a point of comparision for approximations at higher levels of theory (recovery of coupling constant as the model is improved).
Regarding the relationship between the angle formed by the O-H and Cu2O2 plane and the magnetic interaction, which was newly presented this time, the position of the hydrogen atoms is based on calculations, so there is little evidence to say that it is a new discovery.
- The positions of H atoms presented in this study uses the X-ray crystallographic data available from past studies and a clarification has been added to Table 3 and highlighted in yellow. And this correlation for a set of complexes has not been reported to the best knowledge of the authors.
The other mentioned correlation between the 2J value and the calculated result was judged to have no chemical basis to support it.
- The correlation between the experimental and computational values has been reported to clarify that the lower functional and basis sets used to skew the data but not so heavily as to impact further correlation analysis. Two lines have been added to this effect in Computational Methodology and Para 2 of Section 3.2 (highlighted in yellow).
Other correlational analysis have been made to molecular structure and discussed in Section 3.3. Other comparisons, while considered present none or insignificant correlations, were not reported.

Reviewer 3 Report
The present manuscript describes DFT studies on dinuclear hydroxo-bridged copper(II) complexes in order to verify the nature of the magnetic exchange between the metal ions and to estimate the correlation to structural parameters. Various models were used in order to shed light on the role of the ligands, the coordination environment, the spin population and other factors on the exchange coupling constants. Similar studies have been reported earlier, however the present work may add some new aspects on this issue. Therefore, publication is recommended, after minor revision related to 1) in Tables 3-7, the compounds should be given with their formula and not with the CSD reference code, see Tables 1&2; 2) p.14, line 5 from top: Figures 18,19 are missing or wrongly mentioned; 3) caption for Figure 8 is missing; 4) p.11, line 5 from bottom: Table 7 (not Figure 7).
Minor editing of English language is needed in some sentences.
Author Response
- These are corrected.

Round 2
Reviewer 2 Report
This improved version of the paper is worthy of being published in Magnetochemistry as a paper reporting an example of DFT calculation results that express the magneto-structure correlation.